# ECHO: A MULTI-SCALE GRAPH LEARNING FRAMEWORK FOR ENHANCED MOLECULAR REPRESENTATION

## ABSTRACT

Accurate molecular property prediction is a cornerstone of drug discovery. While graph-based models excel at capturing molecular topology, they operate on a single atomic scale, overlooking crucial higher-order information from functional groups and spatial geometry like bond angles. To address this, we propose ECHO, a multi-scale graph learning framework that hierarchically integrates information from atom-level topology, functional-group semantics, and bond-angle geometry. ECHO introduces a hierarchical cross-scale attention mechanism for bidirectional information flow between fine-grained and coarse-grained graph representations, enabling mutual refinement. To further synthesize these diverse features, a novel hypergraph fusion module is designed to capture high-order interactions. Extensive experiments show that ECHO consistently outperforms state-of-the-art baselines, demonstrating the significant advantage of its multi-scale approach and offering new insights into the interplay between different structural scales in molecular representation learning. Code is now available at https://anonymous.4open.science/r/ECHO-3C40.

## 1 INTRODUCTION

With the remarkable enhancement of hardware computing capabilities in recent years, computational methods for predicting molecular properties have witnessed rapid development. In contrast to traditional experimental methods, computational approaches exhibit substantial advantages in terms of high-throughput and efficiency. Machine learning models based on molecule feature engineering have achieved outstanding performance. These models utilize human-defined high-dimensional molecular feature vectors, such as Morgan Fingerprint (MorganFP), as input features(Bonner et al. (2022), Durán et al. (2018), Huang et al. (2020)).

In recent times, deep-learning-based methods, which employ multiple types of chemical or molecular information as input, have started to attract attention due to their state-of-the-art performance in molecular property prediction tasks. Deep-learning-based methods can be categorized into three types. The first type uses sequence information like SMILES (Simplified Molecular-Input Line-Entry System, Weininger (1988))(Zaremba et al. (2014), Goh et al. (2017)). The second category of methods is the traditional style, which utilizes molecular fingerprints, such as Morgan Fingerprint, or other handcrafted features. The third category, end-to-end methods, directly utilizes molecular representations, such as molecular graphs or 3D coordinates, as input for property prediction.

Among them, graph neural network (GNN) methods are becoming more popular, due to their inherent ability to represent molecular structures as graphs. A graph $G(V, E)$ consisting of a node set $V$ and an edge set $E$ can inherently represent molecular structures, where nodes are associated with atoms and edges represent chemical bonds. The graph neural network was initially proposed in 2008(Scarselli et al. (2008)). It can be characterized as a message-passing network among nodes and edges(Xu et al. (2018)), leveraging convolution, self-attention, and other specialized or spectral mechanisms to aggregate information from local or global nodes. Recently, Graph Convolution Network (GCN,Kipf (2016)), Graph Attention Network (GAT, Veličković et al. (2017)), and other graph models have demonstrated significant advantages in modeling social networks, biological relationship networks, and molecular structures. Typically, in end-to-end models, an atom graph is

constructed from SMILES information. For example, (Cereto-Massagué et al. (2015)) takes advantage of graph data.

Although methods that utilize molecular information to construct graphs are widely employed, the limitations of building data solely based on molecular structures are rather evident. Geometric and functional group information is of crucial importance in determining molecular properties. As depicted in Figure 1, an example illustrates how the geometric structure impacts molecular properties. Another instance, similar to the drug pair presented in the figure, is cis - platin and trans - platin(Loehrer & EINHORN (1984)). These two compounds share the same topological structure; however, cis - platin is a highly effective anti - cancer drug, whereas trans - platin is not. In cis - platin, the chlorine atoms are arranged in a cis - configuration as leaving groups. They can simultaneously dissociate two adjacent key bases on the same strand of DNA, thereby forming stable cross - linkages to bind with DNA and exert their effects. Conversely, trans - platin can only bind to a single base on the same strand or to bases on different strands simultaneously. It is unable to form stable bidentate coordination bonds, which makes it difficult for trans - platin to bind to DNA and prevents it from disrupting the DNA of cancer cells.

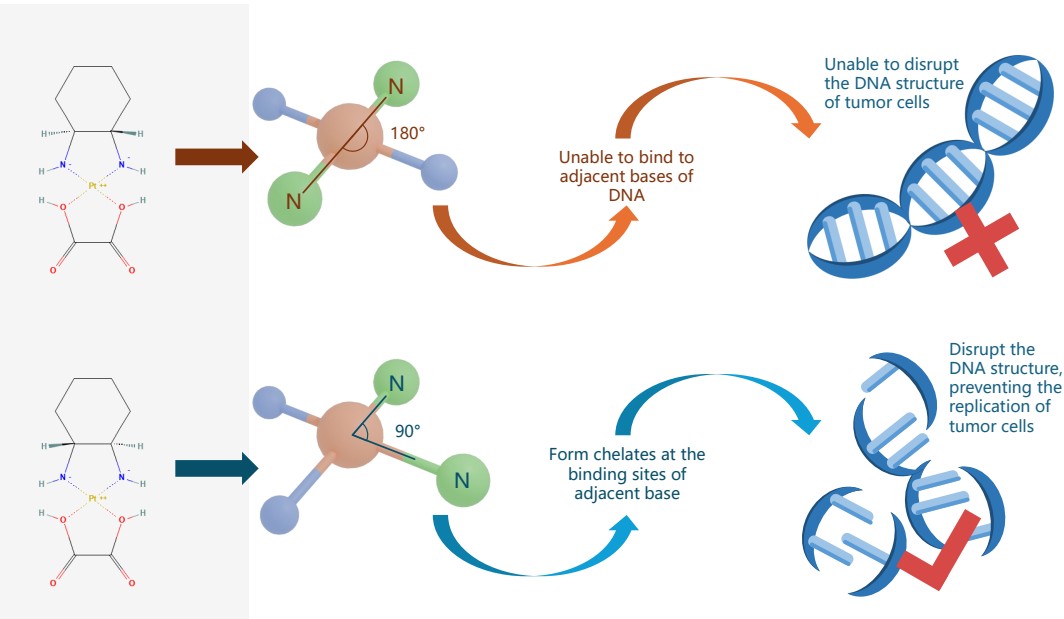

Figure 1: A conceptual example illustrating the critical role of geometric structure in determining molecular properties. The upper row shows a molecule with a 180° bond angle, which is unable to bind to DNA, while the lower row shows a molecule with a 90° bond angle, which can form chelates and disrupt the DNA structure of tumor cells.

Functional group is a set of several atoms, having some special chemical features and usually determining the crucial properties of a molecule. A typical example is the difference between ethanol (CH3CH2OH) and ethane (CH3CH3). Ethanol contains a hydroxyl group, endowing it with polarity. Meanwhile, it is soluble in water because of the hydrogen bond connecting hydroxyls in ethanol and water. Ethanol can also undergo several reactions related to the hydroxyl group. By contrast, ethane is hardly soluble in water and lacks polarity.

Since graphs are non-Euclidean data, in order to maintain the isomorphism of the graph, graph data cannot be added or concatenated like vectors. This makes it difficult to expand graph data to include more chemical information. To some extent, this issue can be alleviated by encoding more information in atomic nodes and chemical bond edges, and by introducing methods such as residual connections. Though some attempts are made to address this issue, due to the over-smoothing characteristic of Graph Neural Networks (GNNs), the advantages of these approaches are not significant.

To address this, we tried to build a heterogeneous graph to construct a more-expressive graph data. A heterogeneous graph $G_H = (V_1, \ldots, V_n; E_1, \ldots, E_n)$ contains different kinds of nodes $V_i$ and

edges $E_i$, where $i$ represents the type of an edge or node. In this work, we fuse geometric information, functional group information, and atom topologic structure together into two heterogeneous graphs $G_{A-F}, G_{A-B}$, while the A-F means atom-functional graph, and the A-B represents the atom-geometric graph, and we build new connections between heterogeneous nodes if they have a relationship, such as a functional group containing several atoms. Thus, the topology structure could represent logical connections of multi-modalities nodes. We also design a novel attention module to aggregate multi-type nodes. It needs to be mentioned that, without geometric and functional group information, our framework will have no difference between using a GAT, so we do not do such ablation experiments.

Another difficulty in fusing multi-modality graph information is designing a high-efficient readout method, because usually molecular property prediction tasks could be conducted as a graph classification or regression procedure, and that needs representation as a vector fed into the fully-connected network or other module. Traditional graph readout methods, including max pooling, global mean pooling and attention pooling(Zhang et al. (2019), Lee et al. (2019)). These methods usually perform well in homogeneous graph pooling, while heterogeneous graph contains multi-type information, making simply calculating the average of node or edge feature lose variant information. To address this, we proposed a cosine-distance based fusion method. It uses the distance between different scale readout vectors to build a hypergraph, and conduct a convolution process to integrate features.

Hypergraph is an advanced form of graph. It is composed of hyperedges and nodes, represented as $H = (V, E)$. Unlike a traditional graph where an edge connects exactly two nodes, a hypergraph allows a hyperedge to connect any number of nodes. The rank, defined as $r(e) = |e|$,, specifies the number of nodes a hyperedge contains. Instead of being defined by the adjacency matrix $\boldsymbol{A}$ like a graph, a hypergraph is defined as an incidence matrix $\boldsymbol{H}$. There exist:

$$A_H = (a_{ij}) \in \{0,1\}^{m \times n},$$

and

$$a_{ij} = \begin{cases} 1, & (v_i \in e_j) \\ 0, & (v_i \notin e_j) \end{cases},$$

where $m$ refers to the number of nodes and $n$ refers to the number of hyperedges.

We proposed several problems and our solutions in previous content. With all these efforts, we established ECHO. The A area of Fig. 2 shows the whole architecture of the ECHO, including several critical modules, which we will introduce in the chapter below. In this chapter, we will illustrate how our model is composed and how it works. For all datasets, we first use RDKit to build RDKit.mol data, and build three scale graph data. We integrate these graphs into two different heterogeneous graphs. Then we proposed an attention-based heterogeneous graph convolution mechanism on them, aggregating information from multi-scale graph nodes. The B area of the Fig. 2 illustrates this mechanism. After this process, we designed a gated fusion module, adding the multi-scale graph readout and morgan fingerprint of the molecule together, and then feed it into a fully-connected layer to get the final result.

Through a series of extensive experiments on benchmark datasets, we demonstrate that ECHO consistently outperforms state-of-the-art baselines, highlighting the significant advantages of our multi-scale and heterogeneous graph learning framework.

## 2 METHODS

### 2.1 THE CONSTRUCTION OF MULTI-SCALE AND HETEROGENEOUS GRAPH DATA

We construct three distinct types of graph data from SMILES information. First, we use RDKit (ref) to transform a SMILES string into a mol object and construct an atom graph, where a node $V_a$ represents an atom and an edge $E_a$ represents a chemical bond. We initialize the original features of an atom node from its specific information, including atom symbol, degree, electronegativity, implicit valence, and aromaticity, using one-hot encoding.

Next, we build a functional group graph from the RDKit mol object and the atom graph, where a node $V_f$ represents a functional group and an edge $E_f$ represents the chemical bond connecting

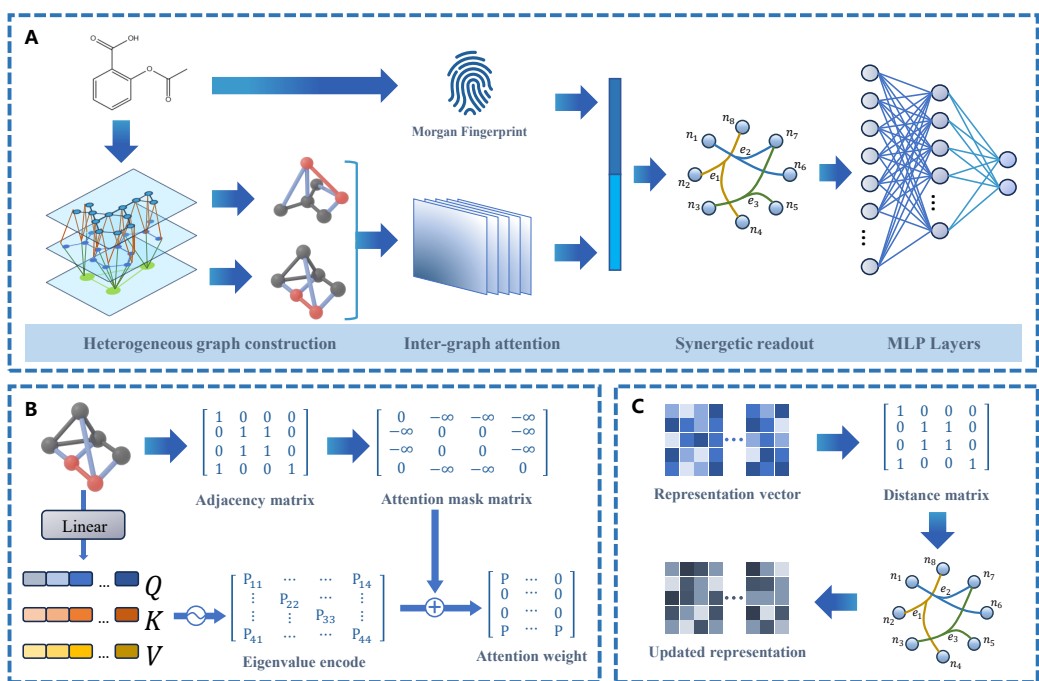

Figure 2: The overall architecture of the ECHO framework, including heterogeneous graph construction, inter-graph attention, synergetic readout, and MLP layers.

functional groups. The featurization of a functional graph node is initialized using its MACCS fingerprint. The MACCS fingerprint is a 166-dimensional vector computed using an asymmetric algorithm. Similar to the Morgan Fingerprint, which is commonly used for molecular featurization, it is uniquely determined by functional groups and contains relevant information about them.

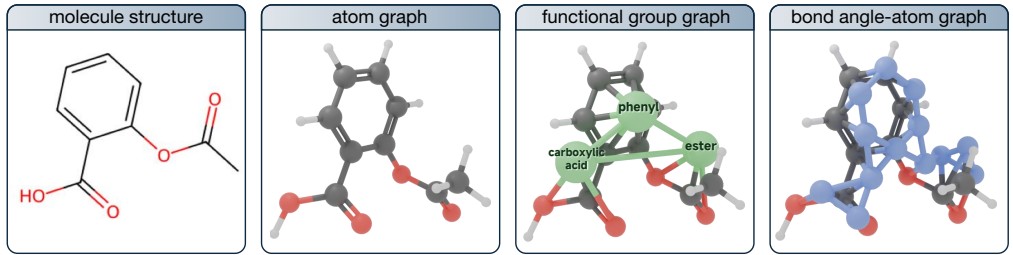

Figure 3: An example of the chemical structure, atom graph, and two heterogeneous graphs composed of the atom graph, functional graph, and bond angle graph, respectively.

The third type of graph is constructed from chemical bond and bond angle information. A node $V_b$ represents a chemical bond and an edge $E_b$ represents a bond angle. We used the MMFF-94 (ref) and UFF force fields provided by RDKit to perform conformational optimization on the 3D structures of molecules, obtaining detailed bond angle data used to encode bond-angle edge features. For chemical bond nodes, we used one-hot encoding based on bond lengths and properties (e.g., single bond, double bond, or others) as features. Additionally, for large molecules that are difficult to optimize with force fields, we adopted a method based on atomic hybridization to estimate bond-angle information and encode the corresponding edge features.

After constructing the three multi-modal graphs, we established a mechanism to integrate the atom graph with the functional graph and bond angle graph into two heterogeneous graphs, as illustrated in Fig. 3. As shown, in the functional group-atom graph, a functional group node is connected to the atom nodes that constitute the functional group. Similarly, a bond node is connected to the atom nodes linked by the bond. We then apply a specialized graph attention mechanism designed to aggregate information from different modalities.

## 2.2 THE MULTI-GRAPH INTERACTIVE ATTENTION

Consider the self-attention mechanism:

$$Attention = softmax(\frac{\boldsymbol{QK^T}}{\sqrt{\boldsymbol{d_n}}})\boldsymbol{V},$$

where the query $\boldsymbol{Q}$, key $\boldsymbol{K}$, and value $\boldsymbol{V}$ are computed from the same input sequence or vector. In this work, we design a heterogeneous graph attention module that calculates $\boldsymbol{Q}$, $\boldsymbol{K}$, and $\boldsymbol{V}$ from different types of nodes. To illustrate, we define $\boldsymbol{X_A} \in \mathbb{R}^{d_A \times 1}$ and $\boldsymbol{X_B} \in \mathbb{R}^{d_B \times 1}$ as feature vectors of different node types, and $\boldsymbol{W_Q}, \boldsymbol{W_K}, \boldsymbol{W_V}$ as linear transformation matrices. The attention can then be expressed as:

$$\boldsymbol{Q} = \boldsymbol{W_Q} \times \boldsymbol{X_A}, \boldsymbol{K} = \boldsymbol{W_K} \times \boldsymbol{X_B}, \boldsymbol{V} = \boldsymbol{W_V} \times \boldsymbol{X_B}.$$

We then apply the same softmax attention with a mask matrix $\boldsymbol{M}$, constructed from the adjacency matrix $\boldsymbol{A}$ of the heterogeneous graph, to transform global attention into local attention, enabling the model to focus on neighboring nodes. To better understand this process, consider the intuitive meaning of query, key, and value. This operation resembles a node of type A asking which nodes of type B contribute most to it. The softmax-based attention uses matrix multiplication with $\boldsymbol{K}$ and $\boldsymbol{V}$ to obtain attention weights of node B relative to node A, achieving a weighted fusion of B with respect to A.

Thus, the multi-graph interactive attention can be expressed as:

$$LocalAtten = softmax[(\frac{\boldsymbol{QK^T}}{\sqrt{\boldsymbol{d_n}}})\boldsymbol{V} + \boldsymbol{M}].$$

This process can also be described for the $k$-th update as:

$$h_{atom}^{(k)} = AGGREGATE^{(k)}(\{h_{funct}^{k-1}, h_{angle}^{k-1}, h_{atom}^{k-1}\}),$$

$$h_{funct}^{(k)} = AGGREGATE^{(k)}(\{h_{funct}^{k-1}, h_{atom}^{k}\}),$$

$$h_{angle}^{(k)} = AGGREGATE^{(k)}(\{h_{angle}^{k-1}, h_{atom}^{k}\}),$$

where $\boldsymbol{h_i}$ denotes the feature of node type $i$. Through this procedure, atom nodes can effectively aggregate information from local atom, functional group, and bond angle nodes.

## 2.3 SYNERGETIC HETEROGENEOUS GRAPH READOUT

Typically, the readout process can be described as:

$$h_G = READOUT(N_i, E_i | i \in G).$$

Traditional readout methods include node feature-based approaches, such as global mean pooling, average pooling, or attention pooling; and topology-based approaches, such as motif-oriented methods that pool the graph into a set of crucial sub-structures called motifs. In our approach, we apply a cosine-distance-based method to readout the heterogeneous graph. We first pool the heterogeneous graph—constructed from atom and functional group nodes $V_A, V_F$ or atom and chemical bond nodes $V_A, V_B$—into node sets $\boldsymbol{N_{A-F}^A}, \boldsymbol{N_{A-F}^F}$, or $\boldsymbol{N_{A-B}^A}, \boldsymbol{N_{A-B}^B}$. The subscript $A - F$ refers to the Atom-Functional group graph, and $A - B$ refers to the Atom-Bond graph.

We compute the cosine-distance matrix among the four different types of graph readout vectors. Then, we traverse the matrix to identify vector sets with a similarity higher than a manually set threshold $\lambda$. If such similarity exists, we define a hyperedge $E_H$ connecting the nodes representing these vectors. Through this process, we obtain the incidence matrix $\boldsymbol{H}$, which uniquely defines a hypergraph $HG$, analogous to how an adjacency matrix defines a graph. Finally, we apply matrix multiplication to the matrix formed by concatenating the four vectors and the hypergraph adjacency matrix $\boldsymbol{A_H}$:

$$N_{fused} = A \times N.$$

This process can also be viewed as a special graph convolution operation since, in the hypergraph $HG$, the vectors represent node features. After multiplication, all features are weighted and fused with their neighbors, with the weight matrix determined by the local structure. We will provide a detailed explanation of this mechanism in the Appendix.

## 3 RESULTS

### 3.1 PERFORMANCE ANALYSIS

We evaluate our model on Tox21, BACE, BBBP, HIV, MUV classification datasets and ESOL, Free-Solv, Lipo regression datasets, choosing evaluation metrics (ROC AUC, RMSE) which are officially recommended by MoleculeNet (Wu et al. (2018)), and the performance of our model and baselines is shown in tables 1 and 2. The ROC AUC means the area under the Receiver Operating Characteristic, and the RMSE means the Root Mean Square Error. The sample and task numbers are also shown in tables. Further introduction about datasets can be found at A appendix chapter.

We choose different kinds of baselines. We first choose GAT, Specformer(Bo et al. (2023)), Graphomer(Ying et al. (2021)), which are relatively classical and modern basic graph neural network models. For the professional molecular representation models, we choose the GeoGNN(Fang et al. (2022)) and Uni-Mol(Zhou et al. (2023)). They are all widely known pre-trained, graph neural network and geometrical information enhanced models.

The ECHO model demonstrates significant advantages in molecular property prediction tasks, especially excelling in classification tasks. On the Tox21 and BACE datasets, its ROC AUC reaches 0.7994 and 0.9202 respectively, both being the highest among all models. In particular, its performance on the BACE task far outpaces that of the second-ranked model, reflecting its strong adaptability to scenarios with moderate sample sizes, focusing on specific targets (such as BACE - 1 enzyme inhibition), or multi - task joint prediction. On the HIV dataset, ECHO ranks first with an ROC AUC of 0.8109, indicating that it can maintain excellent performance in single - task classification with large - scale samples. In regression tasks, ECHO performs best on FreeSolv and ESOL. On the Lipo dataset, it also approaches the optimal with an RMSE of 0.6060, demonstrating good predictive ability for molecular physicochemical properties such as solvation free energy and lipophilicity. It is worth noting that on most of the datasets where it is not the optimal model, the performance of ECHO is only slightly lower than that of large - scale pre - trained models, demonstrating its performance advantages under conditions of low data volume and insufficient computing power.

Table 1: Performance on classification tasks (ROC AUC)

|  | Sample | Task | GAT | Specformer | Graphomer | GeoGNN | Uni-Mol | ECHO |
|---|---|---|---|---|---|---|---|---|
| Tox21 | 7831 | 12 | 0.5870 | 0.7816 | 0.7615 | 0.7810 | 0.7960 | **0.7994** |
| BACE | 1513 | 1 | 0.7075 | 0.8096 | 0.7842 | 0.8561 | 0.8572 | **0.9202** |
| BBBP | 2039 | 1 | **0.7998** | 0.7103 | 0.7919 | 0.7244 | 0.7294 | 0.7313 |
| HIV | 41127 | 1 | 0.7198 | 0.8094 | 0.7642 | 0.8069 | 0.5000 | **0.8109** |
| MUV | 93087 | 17 | 0.6176 | 0.7970 | 0.7664 | **0.8175** | 0.5000 | 0.8000 |

### 3.2 ABLATION ANALYSIS

After removing the molecular fingerprint, the performance of all tasks declined to a certain extent. In classification tasks, the ROC AUC of Tox21, BACE, and BBBP decreased. In regression tasks,

Table 2: Performance on regression tasks (RMSE)

| | Sample | GAT | Specformer | Graphomer | GeoGNN | Uni-Mol | ECHO |
|---|---|---|---|---|---|---|---|
| ESOL | 1128 | 1.5441 | 1.1106 | 0.8773 | 0.7983 | 0.7929 | 0.8601 |
| FreeSolv | 642 | 3.7864 | 2.7642 | 2.0732 | 1.8779 | 1.4804 | 1.3802 |
| Lipo | 4200 | 1.2070 | 0.7393 | 0.7210 | 0.6608 | 0.6033 | 0.6060 |

Table 3: Results of ablation experiments (ROC AUC and RMSE)

| Models | Tox21 | BACE | BBBP | ESOL | FreeSolv |
|---|---|---|---|---|---|
| ECHO w/o MorganFP | 0.7719 | 0.7347 | 0.7110 | 1.5803 | 1.8121 |
| ECHO w/o Synergetic graph readout | 0.7875 | 0.7206 | 0.7240 | 0.9995 | 1.4043 |

the RMSE of ESOL and FreeSolv increased, indicating that the molecular structure - related features provided by MorganFP are important.

Regarding the graph read - out part, after removing this module (ECHO w/o Synergetic graph read-out), the performance was also impaired and the differences among tasks became more prominent. In classification tasks, the ROC AUC of BACE decreased most significantly. In regression tasks, the increase in the RMSE of ESOL was greater than that of FreeSolv. This shows that the synergistic graph read - out module plays a more crucial role in integrating features and improving prediction accuracy in classification tasks focusing on specific targets (such as BACE) and water - solubility prediction (ESOL), while its impact on multi - task classification (Tox21) and solvation free energy prediction (FreeSolv) is relatively limited.

Overall, the absence of either of the two modules will weaken the performance of the ECHO model. Their synergistic effect is an important guarantee for the ECHO model to efficiently complete molecular property prediction tasks.

## REPRODUCIBILITY STATEMENT

We have published all the source code of ECHO on an anonymous GitHub repository. Our model was trained and tested using public datasets. This allows reviewers and readers to reproduce our work, evaluate the performance of ECHO, or apply our approach to their own datasets to further their research.

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

## A APPENDIX

### THE USE OF LARGE LANGUAGE MODELS

In the preparation of this manuscript, a large language model (LLM) was used solely for the purpose of language polishing and refinement, including grammatical corrections and improved phrasing. All scientific content, data, analysis, and conclusions remain the sole responsibility of the authors. This use aligns with ICLR 2026 policies on LLM disclosure.

## A.1 INTRODUCTION OF DATASETS

**Tox21.**The full name of the Tox21 dataset is Toxicology in the 21st Century, which was released in 2014. In the version provided by MoleculeNet, this dataset includes 7831 molecules and 12 toxicity indicators. It is mainly used to evaluate the toxicity of chemical substances on 12 different biological endpoints, such as apoptosis, DNA damage, and hormone disruption. The purpose is to develop and validate computational models for predicting compound toxicity, thereby providing data support for new drug development, environmental monitoring, and the formulation of public health policies.

**BACE.**BACE dataset was published in 2013, including 1513 substances. It provides quantitative (IC50) and qualitative (binary label) binding results for a set of inhibitors of human $\beta$-secretase 1(BACE-1).

**BBBP.**The full name of the BBBP dataset is the Binary labels of blood-brain barrier penetration(permeability). It was published in 2017, providing information on whether 2039 molecules can cross the blood-brain barrier.

**HIV.**A subset of PubChem BioAssay by applying a refined nearest neighbor analysis, designed for validation of virtual screening techniques, provides information about whether 41,127 molecules can have biomedical activity against HIV.

**MUV.**A subset of PubChem BioAssay by applying a refined nearest neighbor analysis, designed for validation of virtual screening techniques, providing information about the activities of 93,087 molecules with multi-type biomedical receptors or targets.

Information about regression dataset is listed. **ESOL.**The ESOL is published in 2004, contains 1128 molecule sampls. It provides Water solubility data(log solubility in mols per litre) for common organic small molecules.

**FreeSolv.**It contains 642 molecules, providing experimental and calculated hydration free energies of small molecules in water.

**Lipophilicity.**It contains 4,200 molecules, providing experimental results of octanol/water distribution coefficient(logD at pH 7.4).

## A.2 HYPERPARAMETER SETTINGS

In the experiment, the hyperparameters of ECHO were set as follows: a single - layer attention layer, two hidden layers of MLP, and eight attention heads. The batch size was set to 32, the number of training epochs was 50, and an early - stopping mechanism was implemented. For other models equipped with the multi - head attention mechanism, the number of attention heads was set to 8. Models with a pre - training component were all trained on their original pre - training datasets. The number of fine - tuning epochs was 20, and an early - stopping mechanism was adopted.

## A.3 THE SYNERGETIC HETEROGENEOUS READOUT

Given input vectors $v_1, v_2, \ldots, v_n \in \mathbb{R}^d$, we first construct the feature matrix $X = [v_1; v_2; \cdots; v_n] \in \mathbb{R}^{n \times d}$.

The key innovation lies in interpreting the cosine distance matrix as a hypergraph incidence matrix:

$$H_{ij} = 1 - \frac{v_i \cdot v_j}{\|v_i\|\|v_j\|}$$

where $H \in \mathbb{R}^{n \times n}$ defines the hypergraph structure, with vectors as nodes and vector pairs as hyperedges.

The adjacency matrix is constructed via matrix multiplication:

$$A = HH^T = \sum_{k=1}^{n} H_{:k}H_{:k}^T$$

with element-wise formulation:

$$A_{ij} = \sum_{k=1}^{n}(1 - \text{sim}(v_i, v_k))(1 - \text{sim}(v_j, v_k))$$

The fusion process $Z = AX$ is equivalent to hypergraph convolution, enabling semantic-aware information propagation. The final output is obtained through global pooling:

$$y = \sum_{i=1}^{n} Z_i = \mathbf{1}^T AX$$

where $\mathbf{1} \in \mathbb{R}^n$ is the all-ones vector.

The method is fully differentiable, with gradients computed as:

$$\frac{\partial Z}{\partial v_i} = \frac{\partial A}{\partial H} \frac{\partial H}{\partial v_i} X + A \frac{\partial X}{\partial v_i}$$

supporting end-to-end training. The weight matrix $A$ automatically achieves semantic awareness: reducing weights between similar vectors while enhancing unique ones, thus optimizing fusion performance.

