# OpenReview forum: "ECHO: A Multi-Scale Graph Learning Framework for Enhanced Molecular Representation"
_ICLR.cc/2026/Conference — Submitted to ICLR 2026_

### Official Review · Reviewer_scR2 · 2025-10-26

**Soundness:** 1
**Presentation:** 1
**Contribution:** 1
**Rating:** 2
**Confidence:** 4

**Summary:**

The paper focuses on enhancing the expressiveness of graph-based models to improve the accuracy of molecular property prediction. The proposed method constructs three types of graphs from molecular structures: an atom graph, a functional group graph, and a bond angle–atom graph, and integrates the atom graph with the other two into heterogeneous graphs. A multi-graph interactive attention mechanism and a readout function are introduced to fuse information across these graphs. Experiments are conducted on both classification and regression datasets.

**Strengths:**

The paper proposes to fuse multiple types of molecular information through heterogeneous graphs to enhance model expressiveness, which is an interesting direction for improving molecular representation learning.

**Weaknesses:**

1. The paper is not well written, which makes it difficult to follow. The problem is not clearly defined, and the description of the proposed method lacks clarity. Moreover, there is no clear statement summarizing the main contributions and limitations of the method.

2. The paper does not provide sufficient comparison with related works, making it difficult to assess the novelty and understand the limitations of existing approaches. In addition, the paper lacks a conclusion section or a clear closing discussion. As a result, the contributions and takeaways are not well summarized, weakening the paper’s overall presentation and clarity.

3. Some statements are vague or unsupported, leading to confusion. For example, in Line 102: “This makes it difficult to expand graph data to include more chemical information. To some extent, this issue can be….” The reasons behind this claim are not provided, and it is unclear how the issue can be alleviated.

4. Some expressions are unsuitable. For instance, in Line 140, the phrase “In this chapter…” should be replaced with “In this section” in the context of a conference paper.

5.  Certain details required for reproducibility and fair evaluation are missing. For example, in Line 215: “We adopted a method based on atomic hybridization to estimate bond-angle information.” The paper does not explain what specific method was used or how it was implemented.

6. Some mathematical symbols are introduced without definition. For example, the meaning of $X_A$ and the rationale for its dimension $d_A \times 1$ are not explained. The use of the multiplication symbol “$\times$” in Line 235 lacks clarification, and symbols and scalars are inconsistently mixed throughout the paper.

7. There are inconsistencies such as “in the Appendix” (Line 281) versus “found at A appendix chapter” (Line 291).

**Questions:**

1. What are the main limitations of current methods that the proposed approach aims to address?

2. In Line 269, what do the four node sets represent? Why are there four sets?

3. Could the authors elaborate on the meaning and roles of $H, A_H$, and $N_{\text{fused}}$?

4. What criteria are used to select the baseline models?

5. Why does the method generate 3D molecular graphs from SMILES strings using RDKit, instead of directly using existing 3D molecular datasets such as QM9 or GEOM-Drug?

---

### Official Review · Reviewer_gsaJ · 2025-10-30

**Soundness:** 2
**Presentation:** 2
**Contribution:** 2
**Rating:** 2
**Confidence:** 4

**Summary:**

This paper proposes a multi-scale graph learning framework named ECHO to enhance the accuracy of molecular property prediction. Its core innovation lies in expanding molecular structures from simple atomic topology graphs to heterogeneous graphs incorporating atomic-level topology, functional group semantics, and bond angle geometry information. ECHO enables information exchange between nodes across scales through cross-scale attention mechanisms. It incorporates a cosine-distance-based hypergraph fusion module for collaborative reading to capture higher-order interactions. Experimental validation across multiple classification (e.g., Tox21, BACE, HIV) and regression (e.g., ESOL, FreeSolv) benchmark datasets demonstrates that Results demonstrate that ECHO consistently outperforms state-of-the-art baselines, with particularly significant advantages in classification tasks for specific targets (e.g., BACE).

**Strengths:**

- By incorporating functional group and bond angle geometric information into molecular graph representations, this approach overcomes the limitations of traditional GNNs that only process atomic topology, offering a novel perspective for molecular representation learning.

- The proposed heterogeneous graph construction, cross-scale attention mechanism, and hypergraph collaborative readout module form a complete and novel technical chain with clear logical coherence.

**Weaknesses:**

1.Although the introduction section mentions the limitations of GNNs (single-atom scale, excessive smoothing), it fails to convincingly demonstrate why "functional groups" and "bond angles" must be introduced and represent the optimal information scales for resolving existing issues. Specifically, the lack of in-depth discussion and comparison with prior work attempting to integrate such information, such as other heterogeneous graph methods[R1], makes it difficult to persuade readers of this work's necessity and uniqueness.

[R1] Zhang et al. Molecular Mechanics-Driven Graph Neural Network with Multiplex Graph for Molecular Structures. Workshop, NeurIPS 2020.

2.The proposed method is common in molecular representation, resembling mechanical stitching, such as multiscale [R1,R2,R3], treating molecular edges as nodes [R4], molecular bond angles [R5], and attention-based readouts [R6].

[R1] Hu et al. MOL-Mamba: Enhancing Molecular Representation with Structural & Electronic Insights. AAAI 2025.

[R2] Yu et al. Self-Supervised Graph Transformer on Large-Scale Molecular Data. NeurIPS 2020.

[R3] Loung et al. Fragment-based Pretraining and Finetuning on Molecular Graphs. NeurIPS 2023.

[R4] Fang et al. Geometry-enhanced molecular representation learning for property prediction. NMI 2022.

[R5] Yue et al. A Plug-and-Play Quaternion Message-Passing Module for Molecular Conformation Representation. AAAI 2024.

[R6] Xiong et al. Pushing the Boundaries of Molecular Representation for Drug Discovery with the Graph Attention Mechanism. Journal of Medicinal Chemistry, 2019.

3.Details regarding functional group-related methods are severely lacking:

    3.1 The distinction between molecular fragments and functional groups: The paper does not explicitly define the specific meaning of "functional group" as used herein. In cheminformatics, a "functional group" refers to a specific chemical moiety (e.g., hydroxyl group -OH) possessing particular chemical properties, whereas "molecular fragments" may broadly encompass any substructure. Does the paper refer to strictly defined functional groups or broader "molecular fragments" obtained through algorithms (e.g., ring-breaking)? This ambiguity in definition directly impacts the reproducibility and comparability of the methodology.

    3.2 Definition, Library, and Number of Functional Groups: The paper mentions "build a functional group graph from the RDKit mol object and the atom graph" but provides no elaboration whatsoever: (1) Which specific functional group definition library was used (e.g., RDKit's built-in, PubChem's, etc.)? Or were custom rules employed? (2) How many distinct functional groups were actually identified and utilized in the model? How does this number impact model complexity and performance?

    3.3 For metal complexes or inorganic compounds (e.g., silicon- or phosphorus-containing molecules), does the functional group definition method described in the paper remain applicable? If not, the model's universality would be severely limited. The paper provides no clarification on this point, constituting a significant omission.

**Questions:**

1.How is the accuracy of automatically identified functional groups from molecules ensured? Are there validation steps? If identification errors occur, how significantly do they impact model performance? These uncertainties remain unevaluated.

2.Why use only MACCS fingerprints? The paper selects MACCS fingerprints for initializing functional group nodes but fails to explain why MACCS was chosen over other fingerprints (e.g., MorganFP, ECFP). As a predefined, substructure-based key-type fingerprint, do its 166 bits optimally represent the semantic meaning of functional groups? Were experiments conducted to compare how different fingerprints characterize functional group nodes? The absence of this justification makes the feature selection appear arbitrary.

3.Dissolution experiments demonstrate performance degradation upon removal of functional group/geometric information, yet the paper merely concludes that "the module is important". The paper fails to deeply analyze why the model performs better with functional groups added. Is it because functional groups themselves provide strong discriminative features, or because heterogeneous graph structures facilitate better message passing? In which specific cases (e.g., distinguishing isomers, identifying active sites) does functional group information play a crucial role? The lack of case studies and attribution makes the advantages of "multiscale fusion" appear vague.

4.Despite ablation experiments, validation of the method's core components remains insufficient. For instance, regarding the collaborative readout module, comparisons were limited to "with" versus "without" scenarios, lacking contrast with more complex readout approaches such as Set Transformers [R1] or other hypergraph pooling methods [R2]. This makes it difficult to demonstrate its superiority.

[R1] Lee et al. Set Transformer: A Framework for Attention-based Permutation-Invariant Neural Networks. ICML 2019.

[R2] Feng et al. Hypergraph Neural Networks. AAAI 2019.

---

### Official Review · Reviewer_pi7p · 2025-10-30

**Soundness:** 2
**Presentation:** 2
**Contribution:** 2
**Rating:** 2
**Confidence:** 4

**Summary:**

This paper introduces a multi-scale grass learning framework for enhanced molecular presentation. The motivation is utilize more chemical information into the representation model. The solution also proposes to construct heterogeneous graphs, including atom-functional graph and atom-geometric graph. Authors also propose a cosine distance based read out method for the heterogeneous graph. Compare to some classical GNN-based method, the method proposing this paper named ECHO shows better performance.

**Strengths:**

The motivation to considering chemical information into molecular representation, learning makes sense and is important.

**Weaknesses:**

1. The existing works related to molecular representation, especially works published in recent two years, and not comprehensively introduced. The main content of the paper lacks a section like Related Works.
2. In line 114-115, the author stated that "without geometric and functional group information, I will framework will have no difference between using a GAT". This statement indicates a limited technical contribution of the proposed method.
3. Also table one and table 2 shows the proposed method performs well on the classification tasks and regression tasks compared to the selected baselines.  Among the compelled to baselines, the most recent work is proposed in 2023, none of the work proposed to in recent years are compared. This does not support the statement in the abstract that "ECHO consistently outperforms state of the art baselines".
4.  More analysis experiment are expected to presented in the experiment section, such as the comparison between existing readout methods  and the proposed readout method.

**Questions:**

Refer to the weaknesses.

---

### Official Review · Reviewer_T8uN · 2025-10-30

**Soundness:** 2
**Presentation:** 2
**Contribution:** 2
**Rating:** 4
**Confidence:** 3

**Summary:**

This paper proposes ECHO, a novel framework for molecular property prediction that addresses a key limitation of GNNs: their operation on a single, atomic scale. The authors argue that higher-order information, such as functional groups and bond-angle geometry, is crucial for accurate prediction but is often overlooked. ECHO hierarchically integrates these three scales (atoms, functional groups, bond angles) by constructing two heterogeneous graphs and introduces two main technical innovations: a cross-scale interactive attention mechanism for information flow between scales, and a synergistic readout module based on cosine-distance hypergraphs to fuse the multi-scale representations effectively. Extensive experiments on MoleculeNet benchmarks demonstrate that ECHO consistently outperforms strong state-of-the-art baselines, including pre-trained models, particularly excelling in classification tasks.

**Strengths:**

1. The core idea of multi-scale molecular representation learning addresses a recognized gap in the field. The motivation is clearly presented, illustrating the importance of geometric and functional information beyond mere topology.

2. The proposed architecture contains several novel components. The ​​heterogeneous graph interactive attention​​ mechanism is a principled approach to enabling bidirectional information flow between different node types.

**Weaknesses:**

1. While the ablation studies on fingerprints and the readout module are valuable, a more critical ablation is missing: ​​the individual contribution of the functional group and bond-angle scales.

2. The paper does not discuss the computational cost and scalability of the ECHO framework.

3. Most baselines are from before 2023, which weakens the persuasiveness of the state-of-the-art (SOTA) claim. Recent methods, such as [1], should be compared.

[1] Advancing Molecular Graph-Text Pre-training via Fine-grained Alignment

**Questions:**

Please see weaknesses

---

### Official Review · Reviewer_m1U2 · 2025-11-03

**Soundness:** 2
**Presentation:** 2
**Contribution:** 2
**Rating:** 4
**Confidence:** 4

**Summary:**

This manuscript proposes a multi-scale heterogeneous graph learning framework called ECHO for molecular property prediction. The method innovatively integrates atom graphs, functional group graphs, and bond-angle graphs, constructs two heterogeneous graph structures, and designs an interactive attention mechanism along with a synergistic readout module, effectively enhancing the expressive capability of molecular graph representations. Experimental results on multiple benchmark datasets demonstrate that ECHO performs strongly in both classification and regression tasks. The main strengths of this work lie in the innovative introduction of multi-scale heterogeneous graph structures and the corresponding information fusion mechanisms. However, the manuscript has significant issues in the methodological description and the experimental design, which undermine the rigor of the manuscript.

**Strengths:**

The core strength of this manuscript lies in its considerable originality. The authors creatively elevate functional groups and chemical bonds to heterogeneous graph nodes, constructing a novel multi-scale molecular representation framework and designing a cross-scale interactive attention mechanism, demonstrating an ingenious conceptual design. This framework exhibits competitive performance across multiple benchmark tests, highlighting the potential significance of this innovative approach. The experimental design is of good quality and generally demonstrates the model’s performance advantages, although there remains room for improvement. Overall, this study has great potential and, if further refined, could be potentially significant.

**Weaknesses:**

The main weaknesses of this paper lie in the severe lack of clarity and critical flaws in quality:

1. Fundamental contradictions in core module descriptions. In the key “synergistic readout” section, the threshold-based discrete hypergraph construction method described in the main text and the distance-based continuous fusion scheme derived in the appendix are two completely different algorithms.
2. Inconsistencies in model architecture description: The manuscript first states that two separate heterogeneous graphs are constructed, yet the subsequent attention update formulas require atom nodes to simultaneously receive information from both functional group and bond-angle nodes, implying a single unified graph structure.
3. Incomplete methodological justification: There is a lack of comparative experiments with the most relevant baseline models, such as methods that enhance multi-scale features on homogeneous graphs, making it impossible to convincingly demonstrate the gains brought specifically by the heterogeneous graph structure rather than by multi-modal information alone.

**Questions:**

1. The manuscript does not sufficiently justify why a heterogeneous graph structure is necessary, rather than simply enhancing a homogeneous graph by incorporating multi-scale information into node features (e.g., concatenating functional group and geometric embeddings). The lack of comparative experiments with such an enhanced homogeneous graph baseline makes it impossible to conclude that the observed performance gains stem from the heterogeneous graph structure itself, rather than merely from the inclusion of multi-modal information. Moreover, the authors’ rationale for not conducting ablation experiments, stating that “without this information, the model is no different from GAT”, is invalid. The proper comparison should involve adding these features to the nodes within a standard GAT.
2. The description of the hypergraph convolution readout method is severely unclear:
(1) The manuscript does not clearly specify whether the “vectors” used to construct the hypergraph are graph-level global representation vectors or the set of embeddings of all nodes. If the former, constructing a hypergraph from only four vectors is overly arbitrary and lacks theoretical justification; if the latter, the authors should explain how the computational complexity arising from a large number of nodes is handled.
(2) The threshold-based discrete hypergraph construction method described in the main text is non-differentiable and cannot support end-to-end training. In contrast, the appendix derivation uses a continuous distance matrix, which is differentiable but deviates from the hypergraph construction logic described in the main text. These two approaches differ fundamentally in both mathematical basis and implementation, constituting a critical contradiction.
(3) The mechanism for choosing the similarity threshold is not explained, and it is unclear whether its manual setting has any theoretical justification. The hypergraph convolution formula lacks a weight matrix and normalization, posing risks of numerical instability. Additionally, the chemical rationale for “connect if similar” is not discussed, leaving open the possibility of introducing invalid edges.
3. Earlier in the manuscript, it is stated that two separate heterogeneous graphs are constructed. However, the subsequent attention formulas require atom nodes to simultaneously receive information from both functional group and bond-angle nodes, implying the existence of a unified heterogeneous graph. This inconsistency between the structural description and the algorithmic design renders the information flow paths completely unclear and must be clarified.
4. Regarding the fusion mechanism for the final model representation, the manuscript exhibits serious logical confusion and missing descriptions. The “gated fusion” module mentioned in the Introduction is completely absent in the Methods section, and the overall workflow depicted in Figure 2 (first performing graph feature fusion, then synergistic readout) fundamentally conflicts with the textual description in the Introduction, which states that the graph readout is added to the Morgan fingerprint.
5. At the end of Chapter 1, the statement “In this chapter, we will illustrate how our model is composed and how it works” appears. However, this content actually belongs to Chapter 2 (METHODS). It is recommended to move it to the corresponding chapter to maintain a rigorous manuscript structure.
6. There is an inconsistency in terminology throughout the manuscript; “readout” and “read-out” are used interchangeably. Please standardize the term.
7. The content at the provided repository link cannot be accessed.

---

### Meta-Review · Area_Chair_H2yw · 2026-01-06

**Summary:**

Reviewers share negative sentiment of this paper. In particular, reviewers are mostly concerned about the following key issues.

- lack of clarity or details, insufficient justification for the proposed method and inconsistencies in the descriptions.

- experiments could be strengthened (ablation study and computational cost)

- more recent related work should be discussed and compared.

In my opinion, these issues are crucial and significant, and must be addressed before publication. However, the authors did not provide any response. Hence, rejection is recommended.

**Reviewer Concerns:**

Authors did not provide rebuttal.

**Reviewer Scores:**

Authors did not provide rebuttal.

---

### Decision · Program_Chairs · 2026-01-26

Reject